# Colorimetric Detection of DNase Type I 3′OH DNA Ends Using an Isothermal Amplification-Assisted Paper-Based Analytical Device

**DOI:** 10.3390/bios12111012

**Published:** 2022-11-13

**Authors:** Wei Xue, Kaiyun Song, Yangyang Chang, Meng Liu

**Affiliations:** Key Laboratory of Industrial Ecology and Environmental Engineering (Ministry of Education) and Dalian POCT Laboratory, School of Environmental Science and Technology, Dalian University of Technology, Dalian 116024, China

**Keywords:** colorimetric, DNA damage, paper-based analytical device, TdT-assisted isothermal amplification

## Abstract

The generation of DNase type I 3′OH DNA ends is closely related to the harm of endogenous reactive oxygen species (ROS) and environmental genotoxic agents. The evaluation of this type of DNA damage plays an important role in clinical intervention and environmental toxicity assessment. Terminal deoxynucleotidyl transferase (TdT)-assisted isothermal amplification (TAIA) offers a facile and versatile way to detect DNase type I 3′OH DNA ends. Its ability of templated-independent isothermal amplification is one unique feature. Here, we reported a paper-based analytical device (PAD) coupled with a smartphone for the detection of DNase type I 3′OH DNA ends using TAIA and colorimetric signal readout. We achieved the integration of cell lysis, DNA extraction, TAIA, horseradish peroxidase (HRP)-enabled colorimetric reaction, and signal readout. This device could achieve a limit of detection of 264 cells with a total assay time of less than 45 min. By combining PAD with a smartphone, the integrated platform could be used for the visual and quantitative analysis of DNA damages with the advantages of ease-to-use, fast response, inexpensive, and instrument free. Furthermore, successful assessment of the genotoxicity in wastewater effluents suggested the great promise of the integrated platform for on-site testing in practical applications.

## 1. Introduction

DNA stores genetic information about the growth and development of all known organisms [1]. The exposure of DNA to endogenous reactive oxygen species (ROS) and environmental genotoxic agents results in strand breaks, modified bases, abasic sites, as well as strand crosslinks [2,3,4]. DNA damage can be genotoxic and has been linked to the development of certain diseases, as well as cell death. Understanding the kind and extent of DNA damage is necessary for developing better clinical therapies and environmental toxicity evaluations.

Nucleic acid amplification, which can generate a large number of target copies and improve the detection sensitivity, is the basis of most molecular diagnostic methods [4,5,6]. Two main approaches including polymerase chain reaction (PCR) and DNA isothermal amplification are commonly used for DNA damage detection. Several approaches have also been developed for the detection of methyl-cytosine or 5-Formylcytosine (5fC) [7,8,9]. However, PCR requires large and expensive precision thermal cyclers, which largely limit its application for on-site testing [10]. To date, efforts have been made to overcome these shortcomings [11,12,13]. Isothermal amplification has emerged as an alternative that can achieve rapid and efficient amplification at a constant temperature without the temperature cycles. Methylation-sensitive restriction endonucleases, recombinase polymerase amplification (RPA), exponential amplification reaction (EXPAR), rolling circle amplification (RCA), or strand displacement amplification (SDA) assisted-systems are currently developed to detect DNA methylation [14,15,16,17]. However, these isothermal amplification approaches still suffer from complicated probe design and specific template-dependent amplification.

A TdT dUTP-mediated nick end labelling (denoted as TUNEL) assay has been employed to detect DNase type I 3′OH DNA ends in vivo and vitro [18]. The mechanism is that TdT has the ability to incorporate biotin-labeled dUTP oligonucleotides to the random DNA strand breaks with 3′ OH ends. Moreover, other types of DNA damage sites, such as oxidative DNA damage, alkylated DNA damage, and deaminated DNA damage, can be converted into 3′ OH ends using corresponding DNA repair enzymes [19,20,21,22]. By combing these enzymes, TdT can be applied for the detection of multiple types of DNA damage [23,24].

Recently, a paper-based analytical device (PAD) coupled with a smartphone have been reported for colorimetric assays, affording a number of affordable, sensitive, specific, user-friendly, rapid and robust, equipment-free, and deliverable to the end user in low-resource settings (ASSURED) tools for on-site testing [25,26]. Integration of a smartphone and PAD device greatly reduces the cost of testing, simplifies the operation process, and minimizes the requirements of the instrument. Various platforms, such as paper chip assays, lateral flow devices, and microfluidic paper-based analytical devices (PAD) have been reported [27,28,29,30]. These devices have been implemented for the visual detection of a wide range of biomarkers.

We describe here an affordable, user-friendly PAD device coupled with a smartphone for quantitative detection of DNA damage. The device integrates on-paper DNA extraction, TAIA, HRP-enabled colorimetric reactions and smartphone imaging. To investigate the feasibility of the device for DNase type I 3′OH DNA ends detection, we compared it with conventional genotoxicity bioassay (comet assay) and TUNEL assay. This device was further used in the genotoxicity detection of pollutants and the evaluation of different wastewater treatment processes for genotoxicity removal. With the merits of rapid, simple, and visual readout, the device offers an ideal platform for on-site genotoxicity evaluation.

## 2. Materials and Methods

### 2.1. Chemicals and Reagents 

Whatman grade 1 chromatography paper (cat. no. 3001-917) was obtained from GE Healthcare (Chicago, IL, USA). The plastic backing pad (cat. no. JY-D101) and adsorbent pad (cat. no. JY-X115) were obtained from JYBIOTECH (Shanghai, China). The 4% paraformaldehyde fix solution (cat. no. E672002-0500) and proteinase K solution (20 mg/mL) (cat. no. B600169-0600) were obtained from Sangon Biotech (Shanghai, China). Terminal deoxynucleotidyl transferase (cat. no. 2230A) was purchased from Takara (Dalian, China). A TUNEL apoptosis detection kit (FITC) (cat. no. C1098) was obtained from Beyotime (Shanghai, China). A TIANAmp genomic DNA kit (cat. no. DP304-03) was obtained from TIAN GEN (Beijing, China). Dulbecco’s modified eagle medium (DMEM) (cat. no. 11965084), 10 × phosphate-buffered saline (PBS) (cat. no. 70011044), fetal bovine serum (FBS) (cat. no. A3160901), and 0.25% trypsin-EDTA (cat. no. 25200072) were obtained from Thermo Scientific (Shanghai, China). A Triton X-100 cell lysis buffer (cat. no. SS0890) was purchased from NOVON (Beijing, China). Deoxyribonuclease I (DNase I) lyophilized powder (cat. no. DN25-10 MG) and all the other chemicals were obtained from Sigma-Aldrich.

### 2.2. Fabrication of PAD

The wax-patterning regions of PAD were designed using Microsoft PowerPoint and printed onto the Whatman grade 1 with a wax printer (Xerox Color Qube 8570). After heating at 120 °C for 1 min, the melted wax could penetrate the paper and form hydrophobic boundaries. Each device contains three pads, a reaction pad, a plastic backing pad, and an adsorbent pad. In total, 5 μL of cell lysis buffer was added onto the reaction pad. After being dried at room temperature, the integrated PAD was assembled as depicted in Figure 1a.

### 2.3. Detection of DNase Type I 3′OH DNA Ends by Standard TUNEL Assay 

The dissociation of Zebrafish liver (ZFL) cells from the primary tissue of zebrafish was described previously [4]. The obtained ZFL cells were fixed with 4% paraformaldehyde for 20 min and then, incubated with 20 µg/mL proteinase K in 100 μL of 1 × phosphate buffer saline (1× PBS, 1.06 mM KH_2_PO_4_, 155.17 mM NaCl, and 2.97 mM Na_2_HPO_4_, pH 7.4) for 10 min. Afterward, the cells were treated with DNase I for 10 min and stained by TUNEL, which labels the damaged cells by incorporating biotin-dUTP to the DNA strand breaks at the 3′OH ends using TdT. The biotin-labeled DNA was then detected by binding to streptavidin–HRP and visualized by 3,3-diaminobenzidine. The brown spots representing damaged cells were imaged using a confocal laser scanning microscope (CLSM).

### 2.4. Colorimetric Detection of DNase Type I 3′OH DNA Ends by PAD 

For generating 3′ OH ends in cells, ZFL cells (~10^6^) were fixed with 4% paraformaldehyde for 20 min and then, incubated with 20 µg/mL proteinase K in 100 μL of 1 × PBS for 10 min. Afterward, the cells were treated with DNase I for 10 min. In total, 10 μL of cell suspensions with various numbers (~ 264, 1320, 6600, 33,000, and 165,000) were applied onto the reaction pad, followed by lysis at room temperature (RT, 25 °C, humidity: 50%) for 5 min. The pad was then washed thrice using 30 μL of 1× PBS. Subsequently, the polymerization reaction was performed in 10 μL of 1× TdT reaction buffer containing 35 nM TdT and 20 µM biotin-dUTP. After incubated at RT for 20 min, 5 μL of 0.5 M EDTA was added to stop the reaction. The paper was washed thrice with 30 μL of 1× PBS to remove the free biotin-dUTP. Then, 10 μL of streptavidin–HRP was added to bind the biotin-labeled DNA. After washing, an equal volume mixture of DAB and H_2_O_2_ is added to the reaction zone for colorimetric reaction. Images were captured using an Honor 9× smartphone (Huawei, Shenzhen, China) based on a homemade mobile phone holder to maintain a focal length constant. ImageJ software was used to analyze the images using a 256-bit color scale. The images were first inverted; thus, the detection zones that were originally white will become black, corresponding to a color intensity of 0 and zones that were originally black will become white, corresponding to a color intensity of 256. Therefore, an increase in the amount of color will result in an increase in color intensity.

### 2.5. Analysis of Genotoxicity of Environmental Pollutants

Different concentrations of ZnO (0–192 µM), K_2_CrO_4_ (0–600 µM), and paraquat (0–960 µM) were added to the culture medium at 37 °C for 4 h. Cells (~10^6^) were washed thrice with PBS, followed by the addition of a cell medium. These cells were then added to the reaction pad as the above-mentioned protocol.

### 2.6. Analysis of Genotoxicity of Wastewater Effluents

Wastewater samples were collected from the AWT process in Hebei Province (China) and stored at 4 °C. The adult zebrafish exposure test was performed according to OECD Guideline 203 (OECD, 1992) [31]. In detail, ten adult zebrafish (with a weight of 0.35 ± 0.06 g) were randomly selected from a stock culture tank and exposed to wastewater samples for 72 h without feeding. The liver tissue was first cut into 3 to 4 mm pieces using sterile scissors. After washing thrice with 1 mL of 1 × PBS, 400 μL of 0.25% trypsin-EDTA was added and incubated at RT for 2 min. The cell suspension was mixed with 800 μL of Dulbecco’s modified eagle medium (DMEM) and filtered through a sterile nylon mesh to separate the cells and tissue fragments. The obtained mixture was then centrifuged at 2000× *g* at 4 °C for 10 min. The purified cells were used in the following study.

### 2.7. Comet Assay

For the detection of DNA damage caused by DNase I, ZFL cells were incubated with proteinase K and DNase I. The supernatant was then removed by centrifugation and the cells were washed thrice with PBS before suspension for further genotoxicity testing. For analysis of the genotoxicity of wastewater effluents, the ZFL cells obtained from the exposed zebrafish were directly used for genotoxicity detection. The genotoxicity detection procedures were performed as a standard comet assay in our previous work [4].

## 3. Results and Discussion

### 3.1. Working Principle of PAD

The paper device consists of three panels, namely, a reaction pad (panel A) for cell lysis, DNA extraction, and DNA damage detection; an adsorbent pad (panel B) for sample purification and washing absorption; and a plastic backing pad (panel C) for sealing the back side of panel A to prevent solution leakage (Figure 1a). Figure 1b shows that the general steps include: (i) cell lysis and DNA extraction. By folding panel B under panel A, ZFL cells are added to panel A preloaded with a dried cell lysis reagent for cell lysis and DNA extraction; (ii) TAIA. 3′OH DNA ends can be incorporated with biotin-dUTP under TdT catalysis; (iii) streptavidin–HRP can bind biotin-labeled DNA; (iv) HRP catalyzes the oxidation of colorless DAB into brown poly (DAB); (v) a smartphone can be used to capture the images; and (vi) followed by data transmission and cloud computing to obtain results on the sites.

### 3.2. Detection of DNase Type I 3′OH DNA Ends by PAD 

The PAD was firstly tested for the detection of DNase type I 3′OH DNA ends. As the levels of 3′OH were typically very low in cells, we first treated the cells with DNase I. A TUNEL reaction (Figure 1a) and comet assay (Appendix A) indicated the 3′OH DNA breaks in DNase I-treated cells. Figure 1b shows that a strong color signal, defined as mean color intensity (MCI), was generated when the following conditions occur: (1) ZFL cells were first treated with DNase I; (2) TAIA reagents including TdT and biotin-dUTP were provided; (3) the streptavidin–HRP was added; and (4) H_2_O_2_ and DAB were provided. These results demonstrated that DNA damage detection could be performed on paper.

### 3.3. Evaluation of the Experimental Parameters

To determine the optimum conditions, we tested different diameters of the circular reaction zone (2, 3, 4, 5, 6, and 7 mm) for colorimetric reaction (Figure 2a). Our results showed that when the outer edge of liquid (>20 µL) does not exceed the reaction zone, the MCI decreases gradually with the increase in the diameter. We also tested the effect of the reaction buffer volume in order to identify the optimal condition for calculating MCI (Figure 2b). The results showed that the use of a 30 µL reaction buffer resulted in the statistically largest color change on the paper chip. However, a larger buffer volume (>30 µL) may cause the outer edge of the liquid to overflow the reaction zone, making the measurement inaccurate. We next optimized the colorimetric reaction time. With increasing the reaction time, the color changed from colorless to brown on the paper surface (Figure 2c). We chose 320 s as the optimal DAB reaction time.

### 3.4. Assay Performance of PAD

We first evaluated the ability of the PAD to detect DNase I induced-DNA damage. Figure 3a shows the relationship between the colorimetric responses and the DNase I treatment time. The colorimetric signal intensity remained unchanged at 12 min. Since exogenous environmental pollutants can easily cause DNA damage, on-site testing of genotoxicity plays an important role in preventing organisms from being harmed. The treated cells with different numbers (~10^2^–10^5^) were added to each reaction zone. As shown in Figure 3b, the PAD provides about three orders of magnitude range for the damaged cells with a limit of detection (LOD) of 264 cells. Therefore, the approach of integrating a smartphone with PAD can realize the simple and quantitative detection of DNA damage.

### 3.5. Detection of DNA Damages Induced by Environmental Genotoxic Agents

To investigate the practical analytical application of the integrated PAD, we employed it to detect DNA damage in the ZFL cells caused by characteristic environmental genotoxic agents, including ZnO, K_2_CrO_4_, and paraquat. These agents have the potential to produce ROS in cells, causing the oxidative stress-related genotoxicity [32,33,34]. Different doses of ZnO (0–192 μM), K_2_CrO_4_ (0–600 μM), and paraquat (0–960 μM) were applied to the ZFL cells for 4 h at 37 °C. Representative CLSM images of these treated cells are shown in Figure 4a, indicating the generation of DNase type I 3′OH DNA ends. Furthermore, it was observed that an obvious signal response was generated with increasing the concentrations of pollutants, yielding a dose-dependent MCI (Figure 4b).

In general, these environmental genotoxic agents could be removed by the advanced wastewater treatment (AWT) processes. Figure 5a shows a typical AWT process, including a secondary sedimentation tank (SST), a tertiary filtration unit (TFU), a membrane unit (MU), and an activated carbon unit (ACU). We first evaluated the genotoxicity removal of AWT by performing the adult zebrafish exposure testing according to OECD Guideline 203 [31]. As shown in Figure 5b, treating zebrafish with SST effluent produced a strong MCI in ZFL cells, suggesting high levels of 3′ OH sites present on genomic DNA. In contrast, MCI signals were almost abolished in the cells from ACU effluent-treated zebrafish. This result demonstrated that ACU could effectively remove exogenous environmental toxicants that can induce DNA damage. The relative levels of 3′ OH sites in the ZFL cells were in accordance with the results in a traditional comet assay (Appendix A and Figure 5c).

## 4. Conclusions

In conclusion, we developed a paper-based device capable of performing cell lysis, DNA extraction, TAIA, HRP-enabled colorimetric reaction, and signal readout. This device could be used for visual and quantitative analysis of DNA damage in cells. The limit-of-detection is 264 cells and the total turnaround time was less than 45 min. For comparison, the Comet assay and TUNEL assay are labor-intensive and laboratory-dependent (Appendix A). This work provides a portable and cost-effective toolset without complicated preparation processes for DNA damage detection on-site. Although we provided an illustrative example for measuring DNase type I 3′OH DNA ends, this approach could be easily extended to other types of DNA damage by combing with various DNA repair enzymes. We envision that this work could serve as the screening tool for genotoxicity testing and environmental health assessment. Future work will develop a 3D-printed device combined with this paper-based device to increase the practicability of on-site detection.

## Data Availability

Not applicable.

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
