# Peer review of "Colorimetric Detection of DNase Type I 3′OH DNA Ends Using an Isothermal Amplification-Assisted Paper-Based Analytical Device"

_biosensors, 2022, doi:10.3390/bios12111012_

Round 1
Reviewer 1 Report
Xue et al. reported a paper-based analytical device for colorimetric detection of DNA damage. In this paper, terminal deoxynucleotidyl transferase-assisted isothermal amplification was used to specifically identify and label the DNase type I 3’OH DNA ends. This detection method can overcome the shortcomings of traditional detection method and meet the on-site detection requirement. This assay has been applied in the genotoxicity assessment of environmental pollutants and wastewater effluents, demonstrating its practicability. I would like to recommend it for publication after addressing my concern, as the novelty is acceptable and the results are important. Please see my comments:
1. The images are not clear, such as scheme 1 and Figure 1. Authors should improve the image quality.
2. In Figure 1, why was the red color uneven on pad? And except positive sample, why did all of pads show the light red color in the presence of HRP?
3. Authors should compare the sensing performance of their method with that of the standard method.
4. The word of “transmision” in Scheme 1 is not correct.
5. I suggest three images of Figure 2 should be put in a row.
6. What is the mechanism of DNA damage caused by these three characteristic environmental genotoxic agents?
7. There are some mistakes in the reference, such as the discrepancy of the title format. Please double check.
Author Response
All the authors would like to express our gratitude to the referees for their constructive comments. Our response to each comment is provided below following each quoted comment.

Reviewer 2 Report
The manuscript by Xue et al. describes a paper-based device for the visual and quantitative analysis of DNA damage. While the results are interesting to the readers of Biosensors, the below concerns would need to be first addressed before further consideration of the manuscript:
The authors are encouraged to proofread their manuscript. There are a few spelling and grammatical mistakes present.
In lines 41-42, authors have to clarify what they mean by their statement or provide suitable references “To date, efforts have been made to overcome these shortcomings”.
In section “2.1 Chemicals and reagents”, the authors are required to provide the purity and concentration of all the chemicals being used in the experiments. Providing any additional identifying information such as manufacturer or part number is also recommended to allow the independent replication of their work. They would also need to clarify if they used Whatman Grade 1 filter or chromatography paper since these have different characteristics.
In section “2.2. Fabrication of PAD”, the authors are required to provide a schematic that shows the different components of the device and the overall dimension of it as well that of the different components. The authors should also provide a picture of the actual physical device.
In line 98 and elsewhere in the manuscript, the authors have to clarify what they mean by 1 × PBS. What’s the molarity or pH of this solution and what’s the volume used in this particular instance?
In line 111, the authors have to clarify if “RT” stands for room temperature and they need to provide the temperature and humidity conditions for the room.
In line 162, the authors mention they used the mean color intensity in their analysis using ImageJ. Does that mean they chose the unweighted greyscale in their RGB color analysis? The authors would need to clarify the color channel they chose in their analysis. Also, from the figures, it seems that the authors have subtracted the measured color intensity value from 255 since the value is increasing with the formation of darker colors. This has to be reflected in the y-axes of the figures. The color produced in the detection zone seems to be red in color which is the complimentary of the blue and green components remaining from the RGB color decomposition in ImageJ. Therefore, the blue or green color channels should show larger change in value over the concentration range tested and this will depend on the peak absorbance of the reaction. The use of this channel should provide more sensitive results:
1. Charbaji et al. Zinculose: A New Fibrous Material with Embedded Zinc Particles. Eng. Sci. Technol. an Int. J. 2021, 24 (2), 571–578. https://doi.org/10.1016/j.jestch.2020.09.005
2. Charbaji et al. A New Paper-Based Microfluidic Device for Improved Detection of Nitrate in Water. Sensors 2021, 21 (1), 1–15. https://doi.org/10.3390/s21010102
3. Kim et al. In Situ Detection of Hydrogen Sulfide in 3D-Cultured, Live Prostate Cancer Cells Using a Paper-Integrated Analytical Device. Chemosensors 2022, 10, 27. https://doi.org/10.3390/chemosensors10010027
In the caption of Figure 1, authors mention that the error bars represent standard deviations; however, they would also need to clarify the sample size. The same applies to the remaining figures in the manuscript that mention the standard deviation without specifying the sample size.
In line 173, what do the authors mean by “paper well”?
In figure 2, the authors have to specify the value of the other variables in each of the different figures such as size, volume and time, etc.
It seems from all the figures, that the sample fluid is large enough to form a bead in the detection zone when photographed. However, in this device, the detection zone is open from top and bottom since it needs to connect to section B and section C at different points in the operation. How is the sample fluid being contained in the detection zone when the user pipettes the sample to form this bead? Wouldn’t it just leak into the table or any surface that touches it due to capillarity and surface tension which results in a loss of sample? The authors are required to show pictures of the actual operation of this device. How does the user pipette the sample while holding the device, connect the different sections and then takes the picture of the detection zone without placing the device on a table or such in order not to lose the sample?
Have the authors considered using a portable 3D printed device for taking the images? This would ensure that background color and light intensity is the same for every photograph independent of environmental variations. Also, the wavelength of the LED in the lightbox can be chosen similar to that of the peak absorbance of the reaction to provide more sensitive results.
Finally, I recommend that the authors improve the quality of the paper by adding more details and clarifications to provide a better understanding of the different sections of the device and the physical operation of it. This will make it easier on the reader to follow and would allow the independent replication of their results. With the above concerns, I can only recommend this manuscript for publication in Biosensors after a major revision.
Author Response

(The authors gave the same response as above.)

Round 2
Reviewer 2 Report
The authors improved the manuscript according to initial comments and recommendations. However, they are encouraged to clarify the following:
Their statement in lines 187-189: “Our results showed that when the outer edge of liquid does not exceed the reaction zone, the MCI decreases gradually with the increase of the diameter.”
Their statement in lines 192-193: “However, larger buffer volume may cause the outer edge of the liquid to overflow the reaction zone, making the measurement inaccurate.” Which volume is that?
Author Response
All the authors would like to express our gratitude to the referees for their constructive comments. Our response to each comment is provided below following each quoted comment.
Reviewer 2 The authors improved the manuscript according to initial comments and recommendations. However, they are encouraged to clarify the following: 1. Their statement in lines 187-189: “Our results showed that when the outer edge of liquid does not exceed the reaction zone, the MCI decreases gradually with the increase of the diameter.” Which volume is that?
Response: We have added the volume: 20 µL, to the revised manuscript.
2. Their statement in lines 192-193: “However, larger buffer volume may cause the outer edge of the liquid to overflow the reaction zone, making the measurement inaccurate.” Which volume is that?
Response: We have added the volume: 30 µL, to the revised manuscript.